# Panel of serum biomarkers for differential diagnosis of idiopathic interstitial lung disease and interstitial lung disease-secondary to systemic autoimmune rheumatic disease

**Miriana d'Alessandro**[1]\*, **Paolo Cameli**[1], **Caroline V. Cotton**[2], **Janine A. Lamb**[3], **Laura Bergantini**[1], **Sara Gangi**[1], **Sarah Sugden**[3], **Lisa G. Spencer**[4], **Bruno Frediani**[5], **Robert P. New**[6], **Hector Chinoy**[6,7], **Elena Bargagli**[1], **Edoardo Conticini**[5]

1 Department of Medical and Surgical Sciences & Neurosciences, Respiratory Diseases Unit, University of Siena, Siena, Italy, 2 Department of Rheumatology, Liverpool University Hospitals NHS Foundation Trust, Liverpool, United Kingdom, 3 Epidemiology and Public Health Group, School of Health Sciences, University of Manchester, Manchester, United Kingdom, 4 Aintree Chest Centre, Aintree Hospital, Liverpool, United Kingdom, 5 Department of Medicine, Surgery & Neurosciences, Rheumatology Unit, University of Siena, Siena, Italy, 6 Faculty of Biology, Medicine and Health, Division of Musculoskeletal and Dermatological Sciences, The University of Manchester, Manchester, United Kingdom, 7 Department of Rheumatology, Salford Royal Hospital, Northern Care Alliance NHS Foundation Trust, Manchester Academic Health Science Centre, Salford, United Kingdom

\* dalessandro.miriana@gmail.com, miriana.dalessandro@unisi.it

**Data Availability Statement:** All relevant data are within the manuscript and its Supporting Information files.

## Abstract

### Background

Interstitial lung disease (ILD) may complicate the course of systemic autoimmune rheumatic disease (SARD) and diagnostic biomarkers are needed. Krebs von den Lungen-6 (KL-6), ferritin (FER) and interleukin 6 (IL-6) have been involved in the ILD development. Our study aimed to compare KL-6, FER, IL-6 and soluble mesothelin-related peptide (SMRP) concentrations in a cohort of idiopathic and SARD-ILD.

### Methods

3169 patients were enrolled in the "UK Biomarkers in Interstitial Lung Disease (UK-BILD) Study". We selected patients affected by SARD-ILD and idiopathic ILD (usual interstitial pneumonia-idiopathic pulmonary fibrosis and fibrotic non-specific interstitial pneumonia). Serum marker concentrations were measured through chemiluminescent assays (Fujirebio Europe, Ghent, Belgium).

### Results

1013 patients were selected for the study: 520 (51.3%) had idiopathic ILD and 493 (48.7%) SARD-ILD. Idiopathic ILD patients displayed higher KL-6 values than SARD-ILD (p = 0.0002). FER and SMRP, though within normal ranges, were significantly higher in idiopathic ILD (p<0.0001). Logistic regression showed good sensitivity (69.4%) and specificity (80.4%) selecting the variables FER and KL-6 concentrations, age and gender-male correlated with a diagnosis of idiopathic ILD.

**Funding:** The author(s) received no specific funding for this work.

**Competing interests:** The authors have declared that no competing interests exist.

**Abbreviations:** ILD, interstitial lung disease; SARD-ILD, ILD associated with systemic autoimmune rheumatic diseases; KL-6, Krebs von den Lungen-6; IL-6, interleukin-6; FER, ferritin; SMRP, soluble mesothelin-related peptide; PFT, pulmonary function tests; HRCT, high-resolution computed tomography; IPF, idiopathic pulmonary fibrosis; DM, dermatomyositis; SSc, systemic sclerosis; RA, rheumatoid arthritis; UK-BILD, UK Biomarkers in Interstitial Lung Disease; pSS primary Sjogren syndrome; SLE, systemic lupus erythematous SLE; MCTD, mixed connective tissue disease; PM, polymyositis; UCTD, undifferentiated connective tissue disease; UIP, usual interstitial pneumonia; NSIP, non-specific interstitial pneumonia; AUROC, areas under the receiver operating characteristic.

## Conclusion

Our study showed the excellent diagnostic value of KL-6 for detecting ILD, which irrespective of the final diagnosis and extent of disease, is always elevated and is a reliable biomarker of lung fibrosis in various diseases, ranging from idiopathic to autoimmune forms. Our study proposed an ILD differentiation model including clinical background. In this context, combination of serum markers and clinical data, as seen in our cohort, may lead to a further improvement in diagnostic accuracy for ILD.

## Introduction

Interstitial lung disease (ILD) may complicate the course of systemic autoimmune rheumatic disease (SARD) [1,2], and is one of the leading causes of morbidity and mortality [3,4]. No definite diagnostic work-up has yet been validated for these patients, nor does robust evidence exist for optimal management, as SARD-ILD not always responds to conventional immunosuppressants, which are the mainstay of therapy for SARD. Diagnosis and management of SARD-ILD requires a comprehensive, multidisciplinary, individualized approach that relies mainly on pulmonary function tests (PFT), high-resolution computed tomography (HRCT) of the chest and sometimes also lung biopsy [5]. A multidisciplinary assessment is highly recommended by international guidelines for the diagnostic pathway of ILD, including at least pulmonologists, radiologists, rheumatologists, in order to optimise the diagnostic accuracy and guarantee the earliest and more proper therapeutic proposal. Despite this recommendation, a significant percentage of ILD patients still receives a "working diagnosis", since clinical symptoms and immunological assessment may not always be sufficient for a confident diagnosis and potentially invasive samplings (such as cryobiopsy or lung surgical biopsy) may not be suitable due to the frailty of clinical conditions [6]. Non-invasive biomarkers for early detection of lung involvement and its severity are badly needed.

Krebs von den Lungen-6 (KL-6) is a high-molecular-weight glycoprotein mainly expressed on proliferating, regenerating and injured type II alveolar epithelial cells (AECs) [7]. It has been suggested as a mainly prognostic serum marker of fibrosis in ILD patients, including those with idiopathic pulmonary fibrosis (IPF) [8,9] and hypersensitivity pneumonitis (HP) [10,11], as well as in SARD-ILD patients, including those with anti-synthetase syndrome (ASS) [12], dermatomyositis (DM) [13,14], systemic sclerosis (SSc) [15–18], primary Sjögren's syndrome (pSS) [19,20], rheumatoid arthritis (RA) [21,22] and ANCA-associated vasculitis [23]. KL-6 concentrations seem to have a positive correlation with the degree of lung impairment detectable by HRCT and a negative correlation with forced vital capacity (FVC) and diffusing capacity of the lungs for carbon monoxide (DLCO) [24]. Since this correlation reflects the severity of SARD-ILD, it can be useful to select patients who could benefit from HRCT and PFT and spare others excessive exposure to radiation and unnecessary procedures, while reducing healthcare costs. Interestingly, in patients with confirmed ILD, KL-6 may decrease during remission of inflammatory activity, but usually remains above normal values.

As far as other serum biomarkers are concerned, ferritin (FER) is a key protein of iron metabolism capable of sequestering large amounts of iron, and thus serves the dual function of iron detoxification and iron storage; it seems to be an important regulator of the immune system, playing a central role in autoimmune diseases [25]. A growing body of data shows that serum FER is correlated with disease activity and poor prognosis in anti-MDA5-positive

DM-ILD patients [26,27], with reported cut-off values that vary from 500 to 1500 ng/ml [28]. Conversely, no data exists on serum FER in other forms of SARD. Interleukin 6 (IL-6) is a pleiotropic cytokine involved in the physiology of virtually every organ system. Controlling IL-6 activity is potentially an effective approach in the treatment of various autoimmune and inflammatory diseases. On the other hand, like calretinin, a well-known marker correlated with IPF severity, soluble mesothelin-related peptide (SMRP) is a surface marker of mesothelial cells, such as pleural mesothelial cells (PMC). No data is available on the role of SMRP in SARD-ILD patients. Nevertheless, recent evidence highlights the role of mesothelin (MSLN which binds cancer antigen CA-125 also known as MUC16) in pulmonary fibrosis, suggesting that MSLN is involved in cell adhesion [29]. The literature reports a role of MUC16 in the development and progression of IPF through the TGF-β1 canonical pathway. The above evidence suggests that FER, KL-6, SMRP and IL-6 may provide a serum biomarker profile that can distinguish the progression of fibrotic damage due to inflammatory activity in SARD-ILD, making it possible to optimize therapeutic management with immunosuppressants and/or antifibrotics.

Here we explore the landscape of serum biomarkers in idiopathic and SARD-ILD in a large cohort of patients from the UK Biomarkers in Interstitial Lung Disease (UK-BILD) Study. The primary endpoint of our study was to assess serum concentrations of IL-6, SMRP, KL-6 and FER in a large cohort of idiopathic or non-idiopathic ILD patients. Secondary endpoints were: to assess whether these biomarkers may be considered specific for SARD-ILD as distinct from idiopathic ILD; to evaluate the association with clinical and imaging findings; and to construct a panel for differential diagnosis.

## Methods

Patients included in the "UK Biomarkers in Interstitial Lung Disease (UK-BILD) Study" were retrospectively enrolled from 39 UK recruitment centres between 07th January 2015 and 07th December 2018. The UK-BILD cohort recruited 3169 in which patients must have HRCT-proven ILD, and their investigations must have included "routine" serology. All recruiting clinicians completed a two-page clinical proforma documenting the following data: age, gender, ethnicity, smoking history, diagnosis of SARD-ILD and idiopathic ILD, SARD signs (including Raynaud, arthralgia/arthritis, sclerodactyly, calcinosis, elevated CK, mechanic's hands, myalgia, periungual erythema, telangiectasia), ILD signs (including digital clubbing, inspiratory crackles, pulmonary hypertension features) and recruiting centre information.

The SARD-ILD group included patients with a diagnosis of pSS, RA, systemic lupus erythematous (SLE), mixed connective tissue disease (MCTD), polymyositis (PM), dermatomyositis (DM), undifferentiated connective tissue disease (UCTD), limited and diffuse SSc and unknown CTD, according to international classification criteria [30–37]. The idiopathic ILD group included patients with a diagnosis of usual interstitial pneumonia-(UIP-)IPF and fibrotic non-specific interstitial pneumonia (NSIP), diagnosed according to American Thoracic Society/European Respiratory Society (ATS/ERS) guidelines [38]. In all Centres, multidisciplinary discussion for diagnostic assessment included respiratory physicians, radiologists, rheumatologists and, in case of tissue sampling for diagnostic purposes, histopathologists, all with a specific expertise in ILD setting. The study was conducted according to the guidelines of the Declaration of Helsinki and approved for "UK Biomarkers in Interstitial Lung Disease (UK-BILD) Study". All patients gave their written informed consent to participation in the study.

For biomarker analysis, inclusion criteria were a diagnosis of IPF, idiopathic NSIP and SARD-ILD; exclusion criteria were a diagnosis of hypersensitivity pneumonitis, sarcoidosis,

asbestosis, idiopathic cryptogenic organizing pneumonia, respiratory bronchiolitis, Langerhans cell's histiocytosis, lymphangioleiomyomatosis, desquamative interstitial pneumonia, acute interstitial pneumonia, lack of serum sample, insufficient or no clinical data, a previous diagnosis (last 5 years) of malignancy and too few patients for statistical analysis.

Serum samples were obtained from recruited patients, anonymized in an electronic database and marker concentrations were measured singly by KL-6, IL-6, FER and SMRP reagent assays (Fujirebio Europe, Ghent, Belgium). The reagents were designed for fully automated chemiluminescent enzyme immunoassay with the LUMIPULSE G System (Fujirebio Europe, Ghent, Belgium). The principle of the assay is agglutination of sialylated carbohydrate antigen with KL-6, IL-6, FER and SMRP mAbs by antigen-antibody reaction. The change in absorbance was measured to determine serum concentrations of KL-6 expressed in U/mL, IL-6 in pg/mL, FER in ng/mL and SMRP in nmol/L. Reference calibrator values were 0 and 1000 ng/ mL for FER, 0, 2 and 100 nmol/L for SMRP, 0, 500 and 10000 U/mL for KL-6 and 0, 20, 400 and 1000 pg/mL for IL-6. The reference intervals for FER concentrations in the low range were 31.5–75.0 ng/mL, and in the high range 280–520 ng/mL. The reference ranges for SMRP were 1.11–1.84 nmol/L (low) and 9.39–15.65 nmol/L (high), and for KL-6 258–387 U/mL (low) and 659–988 U/mL (high). For IL-6 we used the standardized reference ranges 32.2–49.4 pg/mL (low) and 195–299 pg/mL (high) [39].

## Statistical analysis

All data is reported as mean ± standard deviation or median and interquartile range (IQR), as appropriate. The Shapiro-Wilk test was used to determine normal distribution. Multiple comparisons were assessed by non-parametric one-way ANOVA (Kruskal-Wallis test) and the Dunn test. The validity of serum marker concentrations used to distinguish SARD-ILD and idiopathic ILD patients was assessed by areas under the receiver operating characteristic (AUROC) curve. Sensitivity and specificity were calculated for cut-offs of the different variables. The Youden index (J = max [sensitivity + specificity − 1]) was used to establish the best cut-offs.

Patients were further stratified according to HRCT findings and comparative analysis of serum marker concentrations were performed within and between the following groups: the idiopathic ILD group included probable fibrotic NSIP on HRCT (IPF confirmed at multidisciplinary discussion, >65 years old, without UIP confirmation at lung biopsy), definite UIP and definite fibrotic NSIP (UIP confirmed at lung biopsy). For SARD patients, those with RA and SSc displaying a UIP pattern (SARD-UIP) were considered separately from those with NSIP (SARD-NSIP).

Machine learning analysis with variable-importance plot was performed to construct a model selecting variables to make accurate predictions. The more a model relies on a variable to make predictions, the more important it is for the model. Binomial logistic regression and ROC curve analysis were used to predict the diagnostic value of each serum marker/clinical parameter for SARD-ILD against clinical diagnosis. Supervised Principal Component Analysis using Kaiser-Guttman rule was performed in an exploratory approach to identify trends in immunological (KL-6, IL-6, SMRP, FER) and demographic (age) features by 2D representation of the multi-dimensional data set. A p-value less than 0.05 was considered statistically significant. Statistical analysis was performed with GraphPad Prism 9.3 and Jamovi software 2.3.

## Results

The total number of patients selected for the study from UK-BILD cohort was 1239. We excluded 108 (8.7%) from the study due to insufficient or unavailable demographic and clinical

data, 103 (8.3%) due to malignancies, 3 (0.2%) due to inclusion body myositis, 1 due to anti-synthetase syndrome (0.08%) and 11 (0.9%) due to an unknown disease subtype. The remaining 1013 patients (median and interquartile range, 70 (61–77) years) were enrolled in the study: 520 (51.3%) had idiopathic ILD and 493 (48.7%) had been diagnosed with SARD-ILD. Their demographic, clinical and immunological data is reported in Table 1.

## Idiopathic ILD versus SARD-ILD

A higher percentage of older males and former smokers (p<0.0001) was found in the idiopathic ILD group (Table 1). As expected, the two groups showed a clear discrepancy in terms of clinical features. Comparative analysis of serum biomarkers showed higher KL-6 concentrations (Fig 1A) in idiopathic ILD than in SARD-ILD patients (p = 0.0002). Although SMRP and FER concentrations were higher in idiopathic ILD than in SARD-ILD patients (p<0.0001), they remained within normal ranges. IL-6 concentrations were similar in the two groups and were in the normal range. Fig 1B shows the ROC curve to distinguish the two groups on the basis of a SMRP cut-off value of 0.88 nmol/L (sensitivity 52%, specificity 64%), a FER cut-off value of 59.15 ng/mL (sensitivity 54%, specificity 67%) and a KL-6 cut-off value of 1281 U/mL (sensitivity 59%, specificity 54%).

Machine learning analysis with variable-importance plot (Fig 2) was used to select variables to include in the model to obtain accurate predictions. The more a model relies on a variable to make predictions, the more important it is for the model. The variables selected were age, gender-male, FER, SMRP, smoking history, ethnicity-Asian, IL-6, ethnicity-Afro-Caribbean and KL-6: the resulting model showed an accuracy of 0.755 (kappa 0.5087) and AUROC 0.81.

Binomial logistic regression analysis (S1 Table) was performed to understand the effect of demographic (gender, age, ethnicity and smoking history) and immunological (SMRP, FER, KL-6 and IL-6) features on the diagnosis of idiopathic and SARD ILD. The variables most associated with idiopathic ILD were higher concentrations of FER (p = 0.0028) and KL-6 (p = 0.0340), age (p<0.0001) and gender-male (p<0.0001). Higher serum concentrations of FER, KL-6 and IL-6 were recorded in males than females (p<0.0001, p = 0.0014 and p = 0.0209, respectively). The variable ethnicity (Asian and Afro-Caribbean vs Caucasian) was associated with SARD-ILD (p<0.0001). The logistic regression model (Fig 3A) showed an AUROC of 0.832 with best sensitivity (69.4%) and specificity (80.4%).

The supervised Principal Component Analysis plot (Fig 3B) shows how the two groups (idiopathic ILD and SARD-ILD) separated on the basis of selected variables. The first and second components explained 47.4% of the total variance based on immunological and clinical findings showing good clustering for idiopathic ILD and SARD-ILD. The scree plot of Eigenvalues for each principal component was reported in S1 Fig.

According to HRCT stratification, serum markers were compared within and between groups and significant differences were reported in Table 2.

## SARD-ILD subgroup analysis

Patients with SARD were further subdivided according to the specific diagnosis and serum markers, and were compared within and between groups, as well as with idiopathic ILD group (Fig 4).

Finally, patients with SARD were further subdivided according to their signs and symptoms. Those who complained of Raynaud symptoms had lower serum concentrations of SMRP and FER (p<0.0001); arthralgia/arthritis was associated with lower KL-6, SMRP and FER (p = 0.0048, p = 0.0002 and p<0.0001, respectively); sclerodactyly and mechanic's hands with lower FER (p = 0.0192 and p = 0.0364, respectively); elevated CK with lower SMRP, FER

**Table 1. Demographic data, including age, gender, smoking and ethnicity in idiopathic ILD and SARD-ILD groups.**

| | Idiopathic ILD (N = 520) | SARD-ILD (N = 493) | p value |
|---|---|---|---|
| **age** | | | |
| Median (IQR) | 73.5 (68–79) | 64 (54–72) | **<0.0001** |
| **Gender, n (%)** | | | <0.0001 |
| Female | 132.0 (25.4%) | 326.0 (66.1%) | |
| Male | 388.0 (74.6%) | 167.0 (33.9%) | |
| **smoking history, n (%)** | | | <0.0001 |
| Never smoker | 167.0 (32.1%) | 246.0 (49.9%) | |
| Former or current smoker | 353.0 (67.9%) | 247.0 (50.1%) | |
| **ARD subgroup, n (%)** | | | |
| SSc Limited | | 48.0 (9.7%) | |
| SSc Diffuse | | 24.0 (4.9%) | |
| UCTD | | 42.0 (8.5%) | |
| CTD (Unknown) | | 6.0 (1.2%) | |
| MCTD | | 29.0 (5.9%) | |
| PM | | 27.0 (5.5%) | |
| DM | | 26.0 (5.3%) | |
| Sjogren syndrome | | 21.0 (4.3%) | |
| RA | | 198.0 (40.2%) | |
| SLE | | 19.0 (3.9%) | |
| Other | | 53.0 (10.8%) | |
| **Ethnicity, n (%)** | | | <0.0001 |
| Caucasian | 512.0 (98.5%) | 382.0 (77.5%) | |
| Asian | 6.0 (1.2%) | 44.0 (8.9%) | |
| African | 0.0 (0.0%) | 15.0 (3.0%) | |
| Afro-Caribbean | 1.0 (0.2%) | 38.0 (7.7%) | |
| Others–specify | 0.0 (0.0%) | 7.0 (1.4%) | |
| Mixed–specify | 1.0 (0.2%) | 7.0 (0.0%) | |
| **IPF diagnosis, n (%)** | | | |
| Not-UIP | 47.0 (9.0%) | | |
| Definite UIP by HRCT | 332.0 (63.8%) | | |
| Definite Fib NSIP on HRCT and LBx with UIP | 35.0 (6.7%) | | |
| Probable Fib NSIP on HRCT no LBx, >65 yrs old and MDT diagnosis IPF | 106.0 (20.4%) | | |
| **PULMONARY SIGNS/SYMPTOMS** | | | |
| **Clubbing, n (%)** | | | <0.0001 |
| No | 377.0 (72.5%) | 450.0 (91.3%) | |
| Yes | 143.0 (27.5%) | 43.0 (8.7%) | |
| **End inspiratory crackle, n (%)** | | | <0.0001 |
| No | 150.0 (28.8%) | 197.0 (40.0%) | |
| Yes | 370.0 (71.2%) | 296.0 (60.0%) | |
| **Pulmonary hypertension\*, n (%)** | | | 0.1510 |
| No | 508.0 (97.7%) | 475.0 (96.3%) | |
| Yes | 12.0 (2.3%) | 18.0 (3.7%) | |
| **ARD SIGNS/SYMPTOMS:** | | | |
| **Sclerodactyly, n (%)** | | | <0.0001 |
| No | 519.0 (99.8%) | 437.0 (88.6%) | |
| Yes | 1.0 (0.2%) | 56.0 (11.4%) | |
| **Calcinosis, n (%)** | | | <0.0001 |

*(Continued)*

**Table 1.** (Continued)

| | Idiopathic ILD (N = 520) | SARD-ILD (N = 493) | p value |
|---|---|---|---|
| No | 519.0 (99.8%) | 473.0 (95.9%) | |
| Yes | 1.0 (0.2%) | 20.0 (4.1%) | |
| **Raised ck, n (%)** | | | <0.0001 |
| No | 517.0 (99.4%) | 443.0 (89.9%) | |
| Yes | 3.0 (0.6%) | 50.0 (10.1%) | |
| **Mechanic's hand, n (%)** | | | <0.0001 |
| No | 519.0 (99.8%) | 458.0 (92.9%) | |
| Yes | 1.0 (0.2%) | 35.0 (7.1%) | |
| **Myalgia, n (%)** | | | <0.0001 |
| None | 518.0 (99.6%) | 429.0 (87.0%) | |
| Yes | 2.0 (0.4%) | 64.0 (13.0%) | |
| **Periungual erythema, n (%)** | | | 0.0043 |
| No | 492.0 (94.6%) | 440.0 (89.2%) | |
| Yes | 28.0 (5.4%) | 53.0 (10.8%) | |
| **Telangiectasia, n (%)** | | | <0.0001 |
| No | 518.0 (99.6%) | 447.0 (90.7%) | |
| Yes | 2.0 (0.4%) | 46.0 (9.3%) | |
| **Arthralgia/arthritis, n (%)** | | | <0.0001 |
| No | 508.0 (97.7%) | 213.0 (43.2%) | |
| Yes | 12.0 (2.3%) | 280.0 (56.8%) | |
| **Raynaud, n (%)** | | | <0.0001 |
| No | 510.0 (98.1%) | 311.0 (63.1%) | |
| Yes | 10.0 (1.9%) | 182.0 (36.9%) | |
| **Laboratory parameters:** | | | |
| **KL-6 U/mL** | | | |
| Mean (SD) | 1604.4 (1235.2) | 1522.9 (1430.8) | 0.0002 |
| **FER ng/mL** | | | |
| Mean (SD) | 134.3 (138.4) | 87.3 (105.4) | <0.0001 |
| **IL-6 pg/mL** | | | |
| Mean (SD) | 87.8 (222.4) | 99.6 (241.4) | 0.4673 |
| **SMRP nmol/L** | | | |
| Mean (SD) | 1.2 (0.7) | 1.1 (0.7) | <0.0001 |

Clinical findings including ATD subgroups, HRCT patterns and rheumatological signs. Immunological data including serum concentrations of KL-6, FER, IL-6 and SMRP in the ILD and SARD-ILD groups. Abbreviations: ILD, interstitial lung disease; SARD, autoimmune rheumatic disease; RA, rheumatoid arthritis; SLE, systemic lupus erythematous; MCTD, mixed connective tissue disease; UCTD, undifferentiated connective tissue disease; PM, polymyositis; DM, dermatomyositis; SSc, systemic sclerosis; CTD, connective tissue disease; IPF, idiopathic pulmonary fibrosis; UIP, usual interstitial pneumonia; HRCT, high resolution computed tomography; NSIP, non-specific interstitial pneumonia; MDT, multidisciplinary discussion team; LBx, lung biopsy; KL-6, krebs von den lungen-6; FER, ferritin; IL-6, interleukin-6; SMRP, soluble mesothelin-related peptide.

*: Defined as mean pulmonary arterial pressure > 20 mmHg, measured through right heart catheterization.

and IL-6 concentrations (p<0.0001, p = 0.0096 and p = 0.0004, respectively); myalgia with lower SMRP and FER concentrations (p = 0.0032 and p = 0.0191) and periungual erythema with lower SMRP values (p = 0.0110).

Concerning pulmonary signs, patients who showed clubbing showed higher serum concentrations of FER and KL-6 (p = 0.0347 and p = 0.0004, respectively) as well as end inspiratory crackle (p = 0.0279 and p = 0.0002, respectively).

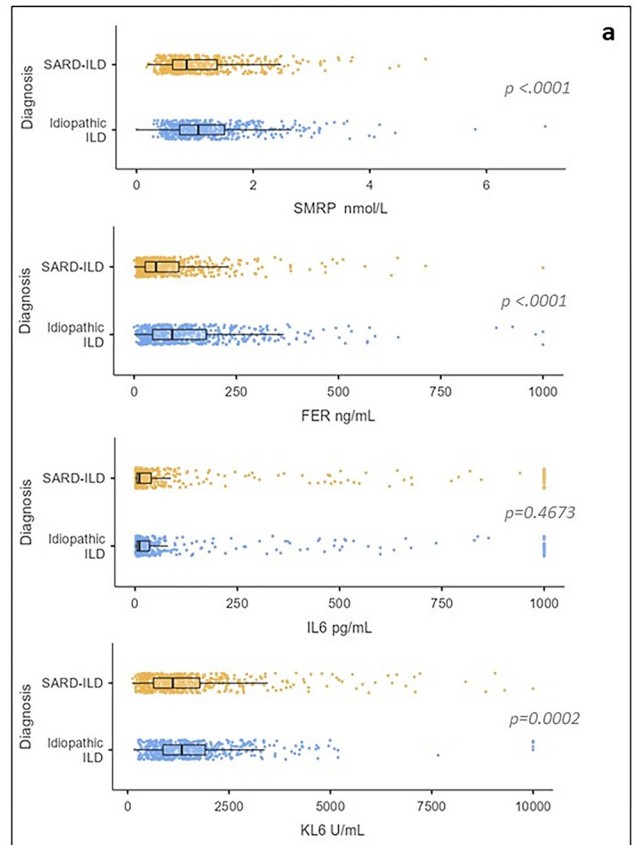

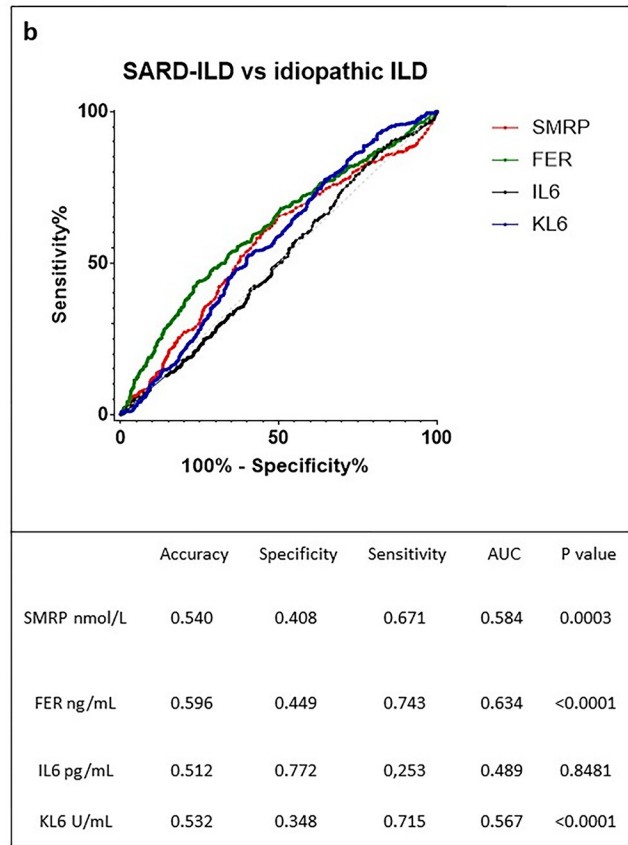

**Fig 1. KL-6, FER, IL-6 and SMRP concentrations in SARD-ILD and idiopathic ILD groups.** Comparative analysis of median concentrations of four markers in the two subgroups (1a) and ROC curve (1b) of serum biomarkers of patients with idiopathic ILD and SARD-ILD reporting specificity, sensitivity, area under the curve and diagnostic accuracy. Abbreviations: KL-6, Krebs von den Lungen-6; FER, ferritin; IL-6, interleukin-6; SMRP, soluble mesothelin-related peptide; ILD, interstitial lung diseases; SARD-ILD, systemic autoimmune rheumatic diseases associated with interstitial lung diseases.

**Table 2. Serum markers concentrations in groups of patients stratified according to HRCT findings: The idiopathic ILD group included probable fibrotic NSIP on HRCT (IPF confirmed at multidisciplinary discussion, >65 years old, without UIP confirmation at lung biopsy), definite UIP and definite fibrotic NSIP (UIP confirmed at lung biopsy).**

| Pairwise comparisons—SMRP nmol/L | | Weight | P values |
|---|---|---|---|
| **S**ARD-NSIP | idiopathic probable fibrotic NSIP | 47.904 | 0.0092 |
| SARD-NSIP | Definite UIP | 52.858 | 0.0026 |
| SARD-UIP | idiopathic definite UIP | -41.369 | 0.0403 |
| **Pairwise comparisons—FER ng/mL** | | | |
| **SARD**-NSIP | idiopathic probable fibrotic NSIP | 63.024 | 0.0001 |
| SARD-NSIP | DefiniteUIP | 67.027 | < .0001 |
| Probable fibrotic NSIP | SARD-UIP | -70.562 | < .0001 |
| Definite UIP | SARD-UIP | -81.872 | < .0001 |
| Definite fibrotic NSIP | SARD-UIP | -40.380 | 0.0493 |
| **Pairwise comparisons—IL6 pg/mL** | | | |
| SARD-NSIP | idiopathic fibrotic NSIP | 5.233 | 0.0030 |
| SARD-NSIP | Definite UIP | 5.750 | 0.0007 |
| SARD-NSIP | SARD-UIP | 6.528 | < .0001 |

*(Continued)*

**Table 2.** (Continued)

| Pairwise comparisons—SMRP nmol/L | | Weight | P values |
|---|---|---|---|
| idiopathic fibroticNSIP | Probable fibrotic NSIP | -4.114 | 0.0422 |
| **Pairwise comparisons—KL6 U/mL** | | | |
| SARD-NSIP | SARD-UIP | -42.924 | 0.0291 |
| Probable fibrotic NSIP | SARD-UIP | -48.549 | 0.0079 |
| Definite UIP | SARD-UIP | -60.183 | 0.0003 |

For SARD patients, those with RA and SSc displaying a UIP pattern (SARD-UIP) were considered separately from those with NSIP (SARD-NSIP). Abbreviations: SARD, autoimmune rheumatic disease; NSIP, non-specific interstitial pneumonia; UIP, usual interstitial pneumonia; KL-6, krebs von den lungen-6; FER, ferritin; IL-6, interleukin-6; SMRP, soluble mesothelin-related peptide.

## Discussion

Our multicentre, retrospective study is the first and largest to evaluate an extended panel of serum biomarkers in patients with different types of ILD, including SARD-ILD. We observed normal serum concentrations of IL-6, FER and SMRP in both groups, whereas KL-6 appeared above normal cut-off in most patients but was significantly higher in the idiopathic ILD group.

These findings, underlining the greater sensitivity and accuracy of KL-6 in the diagnosis of ILD, are not surprising. As early as 2000, Nakajima et al. evaluated serum KL-6 in SARD

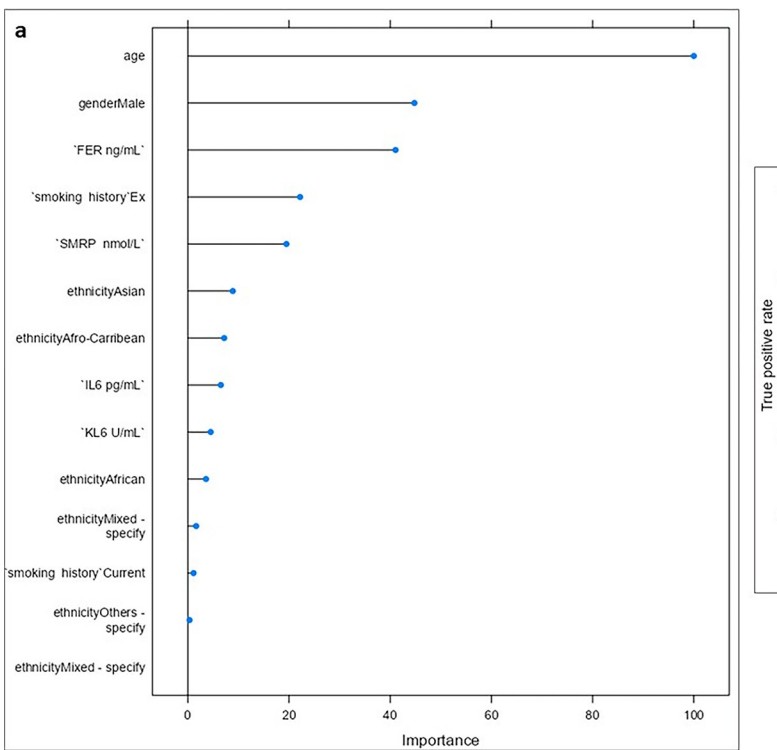

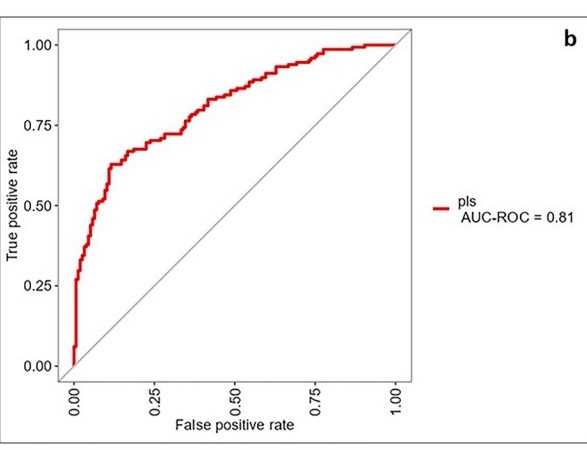

**Fig 2.** Variable-importance plot (a) selecting variables to include in the model to obtain accurate predictions: Age, gender-male, FER, SMRP, smoking history, ethnicity-Asian, IL-6, ethnicity-Afro-Caribbean and KL-6. The area under the receiver operating characteristics (AUC-ROC) curve (b) of the model was 0.81. Abbreviations: KL-6, Krebs von den Lungen-6; FER, ferritin; IL-6, interleukin-6; SMRP, soluble mesothelin-related peptide; ILD, interstitial lung diseases; SARD-ILD, systemic autoimmune rheumatic diseases associated with interstitial lung diseases.

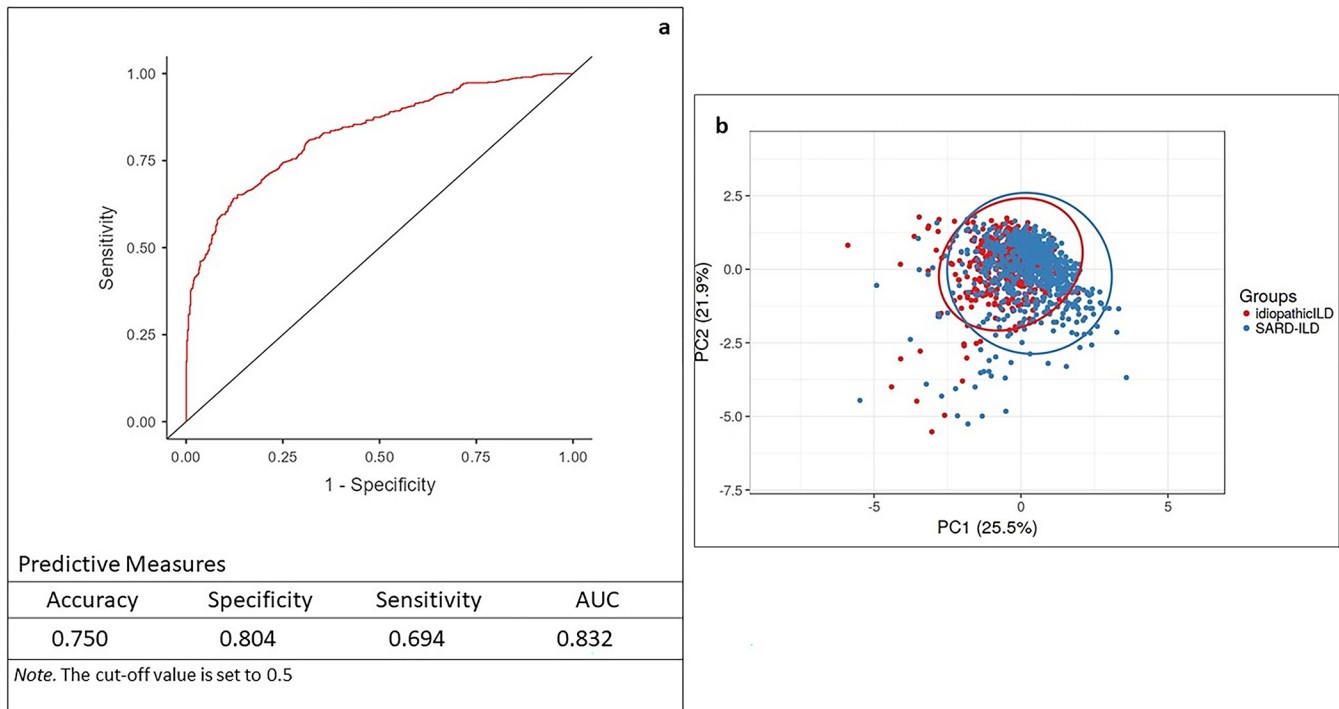

| Predictive Measures | | | |
| --- | --- | --- | --- |
| Accuracy | Specificity | Sensitivity | AUC |
| 0.750 | 0.804 | 0.694 | 0.832 |

*Note.* The cut-off value is set to 0.5

**Fig 3.** Logistic regression model (a) showed an area under the curve (AUC) of 0.832 and an accuracy of 0.75. Principal Component Analysis (b) plot showed that the idiopathic ILD and SARD-ILD groups separated on the basis of the selected variables with a total variance of 47.4%. Abbreviations: PC, principal component; ILD, interstitial lung diseases; SARD-ILD, systemic autoimmune rheumatic diseases associated with interstitial lung diseases.

patients with and without ILD, demonstrating the potential of KL-6 as predictor of lung interstitial involvement and proposing it as a marker of disease activity [40]. Since then, other studies, mainly focusing on SSc, have shown the reliable diagnostic and prognostic value of KL-6 in SARD-ILD: KL-6 seems able to distinguish patients with and without lung involvement at an early stage and shows moderate to high correlations with lung function parameters and quantitative HRCT scores of lung interstitial involvement [41]. The specificity of KL-6 is shown by its capacity to distinguish fibrotic ILD from other types of lung involvement, such as nodular or haemorrhagic pattern in ANCA-associated vasculitis [23].

Ours is the first study to attempt a direct comparison of KL-6 in two groups of ILD. A statistically significant difference was detected: patients suffering from idiopathic ILD displayed higher levels of KL-6 than SARD-ILD patients, suggesting that this biomarker has very high specificity for idiopathic ILD and that different cut-off values are needed for other types of ILD. Likewise FER and SMRP, though within normal ranges, were significantly higher in idiopathic ILD, whereas no statistically significant difference was found for IL-6, which remained within its normal range.

These findings enabled us to build a model with an accuracy of 0.755 for differential diagnosis of idiopathic ILD and SARD-ILD based on the following variables: age, gender-male, FER, SMRP, smoking history, Asian and Afro-Caribbean ethnicity, IL-6 and KL-6. FER and KL-6 concentrations, age and gender-male predicted the diagnosis of idiopathic ILD.

Identification of biomarkers by machine learning classifiers to assist diagnose RA-ILD has been proposed by Qin et al [22]. KL-6 concentration, D-dimer, and tumor markers greatly aided RA-ILD identification. Machine learning algorithms combined with traditional biostatistical analysis could be helpful to diagnose RA-ILD patients and identify biomarkers

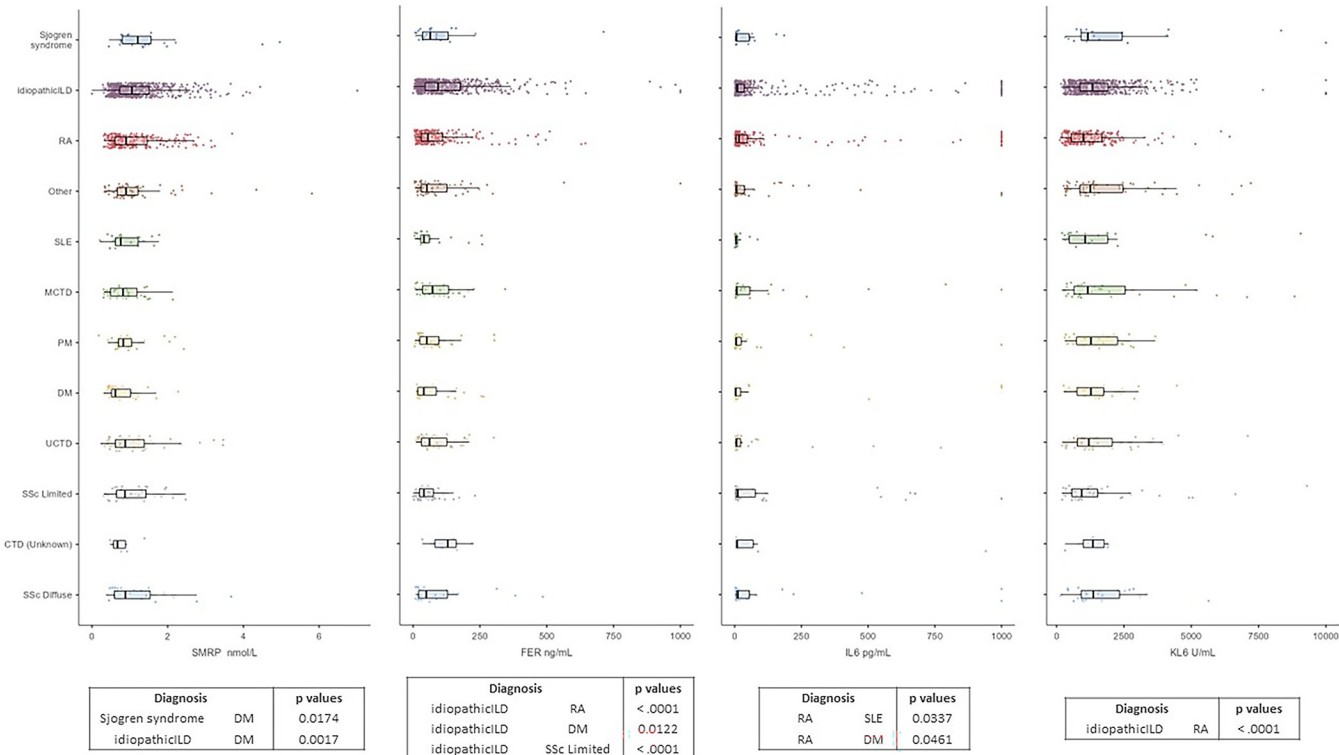

**Fig 4. Box plots reported serum marker concentrations in idiopathic interstitial lung disease (ILD) versus SARD-ILD subgroups: Rheumatoid arthritis (RA), systemic lupus erythematous (SLE), mixed connective tissue disease (MCTD), polymyositis (PM), dermatomyositis (DM), systemic sclerosis (SSc), undifferentiated connective tissue disease (UCTD), connective tissue disease (CTD).** The statistically significant differences of each serum marker concentration between the disease groups were reported in the tables below the box plots.

potentially associated with the disease. Recently, Huang et al [42] proposed multiple machine learning models trained with a large number of proteins involved in the immune pathway consistently distinguished CTD-ILD from IPF in challenging cases and improved clinical decision making.

This is of paramount importance in practice, first because it may help refine and accelerate diagnostic work-up, secondly and more importantly because unnecessary treatment can be avoided and therapy can be targeted.

In order to reduce the risk of bias and to bring our analysis into line with clinical practice, where HRCT has already been performed upon referral, patients were stratified according to radiological pattern. Notably, serum concentrations of FER and SMRP were significantly higher in patients with idiopathic NSIP and UIP than in those with SARD-NSIP and UIP, respectively. At the same time, not only were serum concentrations of KL-6 higher in idiopathic NSIP and UIP patients than in those with SARD-UIP, but also in patients with SARD-NSIP than in those with SARD-UIP.

In a nutshell, elevated levels of KL-6 in a patient with suspected or even radiologically confirmed lung fibrosis are associated with a high probability of ILD. Although FER and SMRP may be in the normal ranges, their increase suggests a diagnosis of idiopathic ILD, and does not support a diagnosis of SARD-ILD. New cut-off values for FER and SMRP, specific for lung fibrosis, could make these biomarkers even more useful in clinical practice.

When we analysed patients with any form of SARD in order to refine panel sensitivity and specificity, we failed to find any statistically significant differences between subgroups, except

in the case of IL-6, which was higher in rheumatoid arthritis patients. This is unsurprising given the pivotal role of this cytokine in the pathogenesis of RA.

On the other hand, interesting insights emerged from clinical findings: patients presenting with signs and symptoms of advanced lung fibrosis (end inspiratory crackles and digital clubbing) had higher serum levels of FER and KL-6, while lower levels of FER, KL-6 and SMRP were recorded in those with extra-pulmonary signs (i.e. arthralgia/arthritis, sclerodactyly, elevated CK, myalgia, periungual erythema, mechanic's hands).

Our study has several limitations: 1) lack of any information about disease activity of the concomitant rheumatic disorders at the time of serum collection; 2) since no lung function data was recorded in UK-BILD, we were unable to compare functional data with serological and imaging findings; 3) autoimmune profile was not included in the proforma: these aspects may have been relevant for stratifying DM subtypes (namely dermatomyositis with anti-MDA5, in which FER is increased) and to refine the diagnosis of many idiopathic NSIP potentially hiding antisynthetase syndrome [43]; such an aspect is worthwhile to be further indagated in upcoming studies; 4) since we lacked a control group of SARD without ILD, we were unable to assess the specificity of FER, SMRP and IL-6.

In conclusion, our study showed the good diagnostic value of KL-6 for detecting ILD, which irrespective of the final diagnosis and extent of disease, seems to be a reliable biomarker of lung fibrosis in various diseases, ranging from idiopathic to autoimmune forms. We confirmed that KL-6 values above 500 U/mL seem to support a diagnosis of ILD in SARD patients (i.e. SSc or IIM-ILD) prior or complementary to HCRT. We also found that assay of serum concentrations of KL-6, combined with FER and SMRP, is useful for differential diagnosis: serum cut-off values of KL-6, FER and SMRP, the latter two within normal values, were validated for differential diagnosis of idiopathic ILD and SARD-ILD. To the best of our knowledge, this is the first time that FER has been thoroughly investigated in patients with lung fibrosis, other than dermatomyositis with anti-MDA5. At the same time, there was no previous data on the role of SMRP in ILD patients and ours is the first study to report higher serum concentrations of SMRP in idiopathic ILD than in SARD-ILD patients, suggesting its potential for differential diagnosis. In this context, combination of serum markers and clinical data, as seen in our cohort, may lead to a further improvement in diagnostic accuracy for ILD.

## Supporting information

**S1 Table. Binomial logistic regression was performed to understand the effect of demographic (gender, age, ethnicity and smoking history) and immunological (SMRP, FER, KL-6 and IL-6) features on the diagnosis of idiopathic ILD and SARD-ILD.** The statistically significant variables was marked in bold (p values columns).
(DOCX)

**S1 Fig. The scree plot method for Kaiser-Guttman's rule to conduct supervised Principal Component Analysis in an exploratory approach for identifying trends in immunological (KL-6, IL-6, SMRP, FER) and demographic (age) features by 2D representation of the multi-dimensional data set.**
(DOCX)

## Acknowledgments

The authors acknowledged FUJIREBIO for the laboratory reagents, and our patients' associations Profondi Respiri and Un Soffio di speranza "Il sogno di Emanuela" ONLUS for their constant help.

## Author Contributions

**Conceptualization:** Miriana d'Alessandro.

**Data curation:** Miriana d'Alessandro, Sara Gangi, Lisa G. Spencer, Robert P. New, Edoardo Conticini.

**Formal analysis:** Miriana d'Alessandro.

**Investigation:** Paolo Cameli, Caroline V. Cotton, Janine A. Lamb, Laura Bergantini, Sarah Sugden, Lisa G. Spencer, Bruno Frediani, Hector Chinoy, Elena Bargagli, Edoardo Conticini.

**Methodology:** Miriana d'Alessandro, Sara Gangi.

**Project administration:** Miriana d'Alessandro, Laura Bergantini, Sara Gangi, Edoardo Conticini.

**Software:** Miriana d'Alessandro.

**Supervision:** Paolo Cameli, Caroline V. Cotton, Janine A. Lamb, Laura Bergantini, Sara Gangi, Sarah Sugden, Lisa G. Spencer, Bruno Frediani, Robert P. New, Hector Chinoy, Elena Bargagli, Edoardo Conticini.

**Validation:** Paolo Cameli, Caroline V. Cotton, Janine A. Lamb, Laura Bergantini, Sara Gangi, Sarah Sugden, Lisa G. Spencer, Bruno Frediani, Robert P. New, Hector Chinoy, Elena Bargagli, Edoardo Conticini.

**Visualization:** Caroline V. Cotton, Janine A. Lamb, Laura Bergantini, Sara Gangi, Sarah Sugden, Lisa G. Spencer, Bruno Frediani, Robert P. New, Hector Chinoy, Elena Bargagli, Edoardo Conticini.

**Writing – original draft:** Miriana d'Alessandro, Paolo Cameli, Caroline V. Cotton, Janine A. Lamb, Sara Gangi, Sarah Sugden, Lisa G. Spencer, Robert P. New, Hector Chinoy, Elena Bargagli, Edoardo Conticini.

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
