## [Decision Letter · Decision Letter 0]

13 Aug 2024

PONE-D-24-29428Panel of serum biomarkers for differential diagnosis of idiopathic interstitial lung disease and interstitial lung disease-secondary to autoimmune rheumatic diseasePLOS ONE

Dear Dr. Dalessandro,

Thank you for submitting your manuscript to PLOS ONE. After careful consideration, we feel that it has merit but does not fully meet PLOS ONE’s publication criteria as it currently stands. Therefore, we invite you to submit a revised version of the manuscript that addresses the points raised during the review process. Specifically, our reviewers found some interests in this manuscript, but also pointed out a number of comments and criticisms. I ask the authors to fully respond to all comments made by reviewers.in the revised version. 

We look forward to receiving your revised manuscript.

Kind regards,

Masataka Kuwana, MD, PhD

Academic Editor

PLOS ONE

Journal Requirements:

Reviewers' comments:

Reviewer's Responses to Questions

**Comments to the Author**

1. Is the manuscript technically sound, and do the data support the conclusions?

Reviewer #1: Partly

Reviewer #2: Yes

2. Has the statistical analysis been performed appropriately and rigorously? 

Reviewer #1: Yes

Reviewer #2: Yes

3. Have the authors made all data underlying the findings in their manuscript fully available?

Reviewer #1: Yes

Reviewer #2: Yes

4. Is the manuscript presented in an intelligible fashion and written in standard English?

Reviewer #1: Yes

Reviewer #2: Yes

5. Review Comments to the Author

Reviewer #1: The is the biomarker study using the large scale of ILD cases. In addition to well-known KL-6, IL-6, and ferritin, a novel marker, SMRP, is investigated as well. Although this is the hot topics of ILD field, I have following concerns.

Major points

[1] The rationale of the study does not seem clear.

Why should the authors use biomarkers to further distinguish between primary and secondary conditions that are already clinically and clearly distinct? It is not clear what is unresolved and why this study is needed.

[2] Abstract and Results

1) “Primary and secondary-ILD”

Define what is “primary ILD” and “secondary ILD”.

2) “ARD”

Systemic autoimmune rheumatic diseases (SARDs) may be better to use.

3) “idiopathic ILD”, “idiopathic UIP”, “idiopathic NSIP”, “ARD-NSIP”, “ARD-UIP”

What kind of guidelines have used these terms? Proper terminology and definitions are needed.

[3] Introduction

Many of the references on the role of KL-6 are published around 2022, but the authors should cite the original paper on the topics. In researching the field, the authors do not appear to have a good grasp of previous research. Reviewing the references once again. Here are the samples.

1) Instead of reference #3, the following paper may help the authors in literature search.

Ishikawa N, Hattori N, Yokoyama A, Kohno N. Respir. Investig. 2012, 50, 3-13.

2) KL-6 was most studied in SSc-ILD. KL-6 was shown to have a prognostic role on the following paper.

Kuwana M, Shirai Y, Takeuchi T. J Rheumatol. 2016 Oct;43(10):1825-1831.

[4] Discussion

The conclusion part is too long and not coherent.

Minor points

[5] Introduction

Ferritin, IL-6, and KL-6 are familiar in the field of ILD. However, SMRP is not.

Explain the reason the authors picked up and investigate it among so many biomarker candidates.

[6] Methods

1) The authors have not mentioned the definition of SSc, MCTD, and UCTD.

2) Mention the definition of “pulmonary hypertension” listed in the Table 1.

3) The results of logistic regression analysis and multivariate analysis were described in the discussion part. Explain them in the methods part.

4) Who examined the HRCT findings?

[7] Figure 1a

We can see 0 and 1 groups. Labeling is required.

[8] Figure 2a

Why did the authors bring IL-6 into the analysis? IL-6 was not shown to distinguish the two groups significantly in Figure 1?

[9] Figure 2a and 4

The letter size of the labeling is too small to see.

Reviewer #2: This paper investigates the utility of KL-6, ferritin, IL-6, and SMRP (soluble mesothelin-related peptide) as biomarkers in distinguishing idiopathic interstitial lung disease (ILD) from ILD associated with autoimmune rheumatic diseases. It proposes a diagnostic model that incorporates clinical background information. Although the study is insightful, the choice of biomarkers, particularly IL-6 and SMRP, appears abrupt and lacks sufficient rationale. The introduction provides a rich explanation of KL-6, but the discussion of ferritin, IL-6, and SMRP in relation to ILD, supported by previous studies, is insufficient. A more robust and logical explanation for selecting these biomarkers is needed. Additionally, the quality of the figures, including resolution and text size, needs improvement to meet the standards of a scholarly article. While the discussion highlights the strengths of the study, it falls short of comparing the findings with previous research, especially concerning the use of machine learning in ILD differentiation and modeling.

Overall, the study's design, the results presented (including the quality of figures), the logical structure of the discussion, and the alignment between the content of the paper and the conclusions emphasized in the abstract are lacking in maturity and refinement. Below are the identified issues:

Major Comments:

1. As mentioned in the overall assessment, the selection of IL-6 and SMRP from the myriad of inflammatory cytokines appears abrupt. The introduction provides a detailed explanation of KL-6, but the discussion of ferritin (FER), IL-6, and SMRP, supported by previous studies, in relation to ILD is insufficient. A clear rationale for selecting IL-6 and SMRP from the numerous ILD-related biomarkers is required.

2. From a clinical application perspective, while specialized tests like IL-6 and SMRP are valuable, the study should also consider practical biomarkers like SP-D and SP-A, which are used in clinical settings alongside KL-6.

3. In the Methods section, if there is a classification criterion for ARD, it should be included. Is there a specific reason for using ARD instead of connective tissue disease (CTD)? The basis for selecting various symptoms and examination findings, such as elevated CK, as ARD signs should be explained with reference to literature.

4. The Methods section lacks citations related to the classification of MCTD and SSc. Anti-synthetase syndrome and UCTD are mentioned in the Results, but their classification criteria should be referenced in the Methods section.

5. The definitions and classifications of chest HRCT findings in ILD should be organized and included in the Methods section, not the Results.

6. The resolution of all figures is poor, and the text is faint and small, making them difficult to read.

7. In Figure 1a, it is unclear whether the vertical axis values (0 or 1) correspond to idiopathic ILD or ARD-ILD.

8. Detailed descriptions of the statistical methods used in Principal Component Analysis are necessary. Was the analysis conducted according to Kaiser-Guttman’s rule using the scree plot method? A scree plot should be added as a supplementary figure. How were the absolute variables selected?

9. In Table 2, why does the number of diseases compared vary for each serum biomarker? What does "W" stand for?

10. In Figure 4, the diseases compared are unclear. It seems that only statistically significant differences are presented, but the rationale is not provided, making it difficult to understand. The figure legends are also insufficient.

11. The discussion focuses on emphasizing the strengths of the study's findings, but lacks a comparison with previous research, especially regarding ILD differentiation using machine learning and related models.

12. In the Abstract’s discussion, the focus should be on proposing an "ILD differentiation model including clinical background," but instead, the emphasis seems to be on the utility of serum SMRP levels.

Minor Comments:

13. In Table 1, the ARD subgroup should be arranged in the order of SSc Limited, SSc Diffuse, UCTD, and CTD (Unknown).

14. In the Binomial logistic regression analysis (Table S1), it would be less confusing for readers if the comparison targets were labeled as idiopathic ILD and ARD-ILD rather than primary and secondary ILD.

6. PLOS authors have the option to publish the peer review history of their article (what does this mean?). If published, this will include your full peer review and any attached files.

Reviewer #1: No

Reviewer #2: No

---

## [Author Response · Author response to Decision Letter 0]

3 Sep 2024

Dear Editor,

The authors would like to thank you and appreciate the Reviewer for the constructive comments. We also thank the Reviewer for the effort and time put into the review of the manuscript. The comments are encouraging, and the Reviewer appears to share our judgement that this study and its results are important and worth publication. Each comment has been carefully considered point by point and responded to. Please see below our detailed response to the comments raised, which you can find in italic. Changes provided in the revised manuscript are in red. The manuscript has been revised according to the Reviewer’s suggestions and has been significantly improved. We are hoping that its final version deserves publication in “Plos One”.

Reviewer #1: The is the biomarker study using the large scale of ILD cases. In addition to well-known KL-6, IL-6, and ferritin, a novel marker, SMRP, is investigated as well. Although this is the hot topics of ILD field, I have following concerns.

Major points

[1] The rationale of the study does not seem clear.

Why should the authors use biomarkers to further distinguish between primary and secondary conditions that are already clinically and clearly distinct? It is not clear what is unresolved and why this study is needed.

Thanks for giving us the chance to clarify such a crucial aspect of our study. In the common clinical practice, clearly distinguishing between primary and secondary ILD is not always easy. For instance, it is not so uncommon to have patients who were previously diagnosed with idiopathic NSIP and then display a positivity of any MSA. This is, according to us, a major point, because while idiopathic ILD may benefit from antifibrotic agents, secondary ones may require a prompt immunosuppressive treatment. Indeed, a multidisciplinary assessment is highly recommended by international guidelines for the diagnostic pathway of ILD, including at least pulmonologists, radiologists, rheumatologists, in order to optimise the diagnostic accuracy and guarantee the earliest and more proper therapeutic proposal. It is worthy to consider that, despite this recommendation, a significant percentage of ILD patients still receives a “working diagnosis”, since clinical symptoms and immunological assessment may not always be sufficient for a confident diagnosis and potentially invasive samplings (such as cryobiopsy or lung surgical biopsy) may not be suitable due to the frailty of clinical conditions (Am J Respir Crit Care Med. 2019 Nov 1;200(9):1146-1153. doi: 10.1164/rccm.201903-0493OC.). For this reason, the validation and discovery of non-invasive and reproducible biomarkers that may help to improve our diagnostic accuracy is highly needed. 

In this setting, in order to validate our findings and our proposed biomarkers, we needed a cohort of patients in whom a definite diagnosis was already performed.

We added such finding in the introduction section of revised manuscript as follows:

“A multidisciplinary assessment is highly recommended by international guidelines for the diagnostic pathway of ILD, including at least pulmonologists, radiologists, rheumatologists, in order to optimise the diagnostic accuracy and guarantee the earliest and more proper therapeutic proposal. Despite this recommendation, a significant percentage of ILD patients still receives a “working diagnosis”, since clinical symptoms and immunological assessment may not always be sufficient for a confident diagnosis and potentially invasive samplings (such as cryobiopsy or lung surgical biopsy) may not be suitable due to the frailty of clinical conditions (Am J Respir Crit Care Med. 2019 Nov 1;200(9):1146-1153. doi: 10.1164/rccm.201903-0493OC.)”

[2] Abstract and Results

1) “Primary and secondary-ILD”

Define what is “primary ILD” and “secondary ILD”.

Thanks for evidencing this mistake. “Primary” and “secondary” ILD were changed into “idiopathic” and “SARD”: we hope that the manuscript could look now more homogeneous in terminology

2) “ARD”

Systemic autoimmune rheumatic diseases (SARDs) may be better to use.

The definition has been modified accordingly

3) “idiopathic ILD”, “idiopathic UIP”, “idiopathic NSIP”, “ARD-NSIP”, “ARD-UIP”

What kind of guidelines have used these terms? Proper terminology and definitions are needed.

Thanks for raising this point. pSS, RA, SLE, MCTD, SSc and IIM were diagnosed according to currently available classification criteria (see ref. 24-27), while idiopathic ILD according to American Thoracic Society/European Respiratory Society (ATS/ERS) guidelines (ref. 28).

[3] Introduction

Many of the references on the role of KL-6 are published around 2022, but the authors should cite the original paper on the topics. In researching the field, the authors do not appear to have a good grasp of previous research. Reviewing the references once again. Here are the samples.

1) Instead of reference #3, the following paper may help the authors in literature search.

Ishikawa N, Hattori N, Yokoyama A, Kohno N. Respir. Investig. 2012, 50, 3-13.

2) KL-6 was most studied in SSc-ILD. KL-6 was shown to have a prognostic role on the following paper.

Kuwana M, Shirai Y, Takeuchi T. J Rheumatol. 2016 Oct;43(10):1825-1831.

Thank you for the suggestion, we modified it accordingly.

[4] Discussion

The conclusion part is too long and not coherent.

Thanks for the suggestion: the conclusion has been shortened to a few bullet points which are more useful for the comprehension of the manuscript.

Minor points

[5] Introduction

Ferritin, IL-6, and KL-6 are familiar in the field of ILD. However, SMRP is not.

Explain the reason the authors picked up and investigate it among so many biomarker candidates.

Thanks for raising this important point: as correctly stated, no study has to date evaluated the role of SMRP in ILD. Nevertheless, recent evidence highlights the role of mesothelin (MSLN which binds cancer antigen CA-125 also known as MUC16) in pulmonary fibrosis, suggesting that MSLN is involved in cell adhesion [Rump et al., 2004]. The literature also reports a role of MUC16 in the development and progression of IPF through the TGF-β1 canonical pathway. In this regard, taking action from these preliminary findings, we decided to assess whether SMRP, aside from “classical” biomarkers, could give interesting insights in this field. Such a statement has been added in “Introduction” section.

[6] Methods

1) The authors have not mentioned the definition of SSc, MCTD, and UCTD.

Thank you. The classification criteria of the abovementioned criteria has been added, accordingly.

New references were added in the revised manuscript: 

SSC: 24122180

MCTD: R. Kasukawa, T. Too, S. Miyawaki, H. Yoshida, K. Tanimoto, M. Nobunaga, et al. Preliminary diagnostic criteria for classification of mixed connective tissue disease 

UCTD: “Due to the lack of validated diagnostic or classification criteria, we employed the preliminary ones relased in 1999 (10544849)

ASS: 21138882

2) Mention the definition of “pulmonary hypertension” listed in the Table 1.

Thanks for your comment. According to your advice, we have added the definition of pulmonary hypertension as defined by international guidelines, that we apply for our patients in the multidisciplinary evaluation according to our Centre protocol. 

3) The results of logistic regression analysis and multivariate analysis were described in the discussion part. Explain them in the methods part.

Thank you for the comment. We moved the results of logistic regression analysis and multivariate analysis in the results section. 

4) Who examined the HRCT findings?

Thanks for your comment that helped us to further clarify this aspect for the readers. In the Centres included in the study, all patients with a clinical suspect or diagnosis of ILD undergo a specific diagnostic pathway, including a multidisciplinary discussion to optimise the diagnostic accuracy, according to international guidelines. Therefore, HRCT images are performed and reviewed by radiologists with a specific expertise on this field, that are part of the multidisciplinary group. 

To better clarify this point, we have added a specific sentence in the Methods’ section. 

[7] Figure 1a

We can see 0 and 1 groups. Labeling is required.

Thanks for the comment. We modified it accordingly. Moreover, we replaced “SARD-ILD” instead of “ARD-ILD” as suggested above. 

[8] Figure 2a

Why did the authors bring IL-6 into the analysis? IL-6 was not shown to distinguish the two groups significantly in Figure 1?

Thank you for the comment. We performed the unsupervised machine learning analysis with variable importance plot to construct a model selecting variables to make accurate predictions. In figure 2a we reported all variables, though not statistically significant in the comparative analysis, to demonstrate that IL-6 did not affect the diagnosis SARD-ILD vs Idiopathic ILD. Here is reported the variable importance plot without IL-6 concentrations.

[9] Figure 2a and 4

The letter size of the labeling is too small to see.

Thank you for the comment. We modified the figures accordingly.

Reviewer #2: This paper investigates the utility of KL-6, ferritin, IL-6, and SMRP (soluble mesothelin-related peptide) as biomarkers in distinguishing idiopathic interstitial lung disease (ILD) from ILD associated with autoimmune rheumatic diseases. It proposes a diagnostic model that incorporates clinical background information. Although the study is insightful, the choice of biomarkers, particularly IL-6 and SMRP, appears abrupt and lacks sufficient rationale. The introduction provides a rich explanation of KL-6, but the discussion of ferritin, IL-6, and SMRP in relation to ILD, supported by previous studies, is insufficient. A more robust and logical explanation for selecting these biomarkers is needed. Additionally, the quality of the figures, including resolution and text size, needs improvement to meet the standards of a scholarly article. While the discussion highlights the strengths of the study, it falls short of comparing the findings with previous research, especially concerning the use of machine learning in ILD differentiation and modeling.

Overall, the study's design, the results presented (including the quality of figures), the logical structure of the discussion, and the alignment between the content of the paper and the conclusions emphasized in the abstract are lacking in maturity and refinement. Below are the identified issues:

Major Comments:

1. As mentioned in the overall assessment, the selection of IL-6 and SMRP from the myriad of inflammatory cytokines appears abrupt. The introduction provides a detailed explanation of KL-6, but the discussion of ferritin (FER), IL-6, and SMRP, supported by previous studies, in relation to ILD is insufficient. A clear rationale for selecting IL-6 and SMRP from the numerous ILD-related biomarkers is required.

Thanks for raising this important point: as correctly stated, no study has to date evaluated the role of SMRP in ILD. Nevertheless, recent evidence highlights the role of mesothelin (MSLN which binds cancer antigen CA-125 also known as MUC16) in pulmonary fibrosis, suggesting that MSLN is involved in cell adhesion [Rump et al., 2004]. The literature also reports a role of MUC16 in the development and progression of IPF through the TGF-β1 canonical pathway. In this regard, taking action from these preliminary findings, we decided to assess whether SMRP, aside from “classical” biomarkers, could give interesting insights in this field. Such a statement has been added in “Introduction” section.

Interleukin-6 (IL-6) is a pleiotropic cytokine involved in the physiology of virtually every organ system. Controlling IL-6 activity is potentially an effective approach in the treatment of various autoimmune and inflammatory diseases. The recent introduction of tocilizumab, a humanised monoclonal antibody targeting IL-6R, is further evidence of the role of IL-6 in the pathogenesis of RA. Baricitinib is a JAK inhibitor that blocks intracellular signalling pathways of inflammatory cytokines recommended for Rheumatoid arthritis (RA) patients not responding to initial treatment. Baricitinib was demonstrated to be a safe immune modulator that reduces the concentrations of biomarkers of lung fibrosis and inflammation in RA patients, including a subgroup with interstitial lung involvement. 

As a result, we included IL-6 and SMRP concentrations in different SARD-ILD compared with idiopathic ILD. 

2. From a clinical application perspective, while specialized tests like IL-6 and SMRP are valuable, the study should also consider practical biomarkers like SP-D and SP-A, which are used in clinical settings alongside KL-6.

Thank you for the suggestion. We selected four markers detectable at the same time using a few microliters (about 150uL) of serum through the chemiluminescence method using Fujirebio reagents. We included KL-6, ferritin and IL-6 as valuable markers in the contest of idiopathic interstitial lung diseases and in a few types of SARD-ILD (e.g. RA).

To date no study has evaluated the role of SMRP in ILD. Nevertheless, recent evidence highlights the role of mesothelin (MSLN which binds cancer antigen CA-125 also known as MUC16) in pulmonary fibrosis, suggesting that MSLN is involved in cell adhesion. An epithelial cell's apical surface is covered with transmembrane mucins, which are large glycoproteins. Cell surface mucins are characterized by non-covalent sodium dodecyl sulfate-labile bonds holding dimerizations of two dissimilar subunits together (α and β chains). It is highly glycosylated and entirely extracellular. High glycosylation levels in this extracellular domain contribute to barrier formation in addition to protecting the protein backbone from proteolytic attacks by hosts. In addition to the putative phosphorylation sites present in all transmembrane mucins intracellular tails, these tails differ in sequence and length and do not contain conserved domains. It is nevertheless believed that transmembrane mucins contribute significantly to cellular proliferation, apoptosis, and epithelial to mesenchymal transition processes, in accordance with IPF observations. According to the above information, the lung is predominantly composed of MUC1, MUC4, and MUC16 TM mucins. 

 3. In the Methods section, if there is a classification criterion for ARD, it should be included. Is there a specific reason for using ARD instead of connective tissue disease (CTD)? The basis for selecting various symptoms and examination findings, such as elevated CK, as ARD signs should be explained with reference to literature.

Thanks for raising this important point. All classification criteria were added (unfortunately, some of them were not mentioned in the first version of the manuscript) and can now be found among the references. In terms of inclusion criteria, we reckon that the choice of including not only CTD but ARD “in general” can not easily understandable and deserves some clarifications: first, the occurrence of ILD is not limited to CTD, but has been reported in up to 10% of patients with RA and its precocious diagnosis and management remain an authentic dilemma. Secondly, the most common pattern of ILD in RA is UIP, which, in turn, is usually present only in SSc among CTDs, which generally have a NSIP pattern. At the same time, the pathological mechanisms of RA, driven also by IL-6 (which is less prominent in SSc, as proved by the uncertain efficacy of TCZ in this condition), could potentially aid in evaluating differences and similarities of ILD development in such diseases. Same when comparing RA with IIM and other CTDs, in which the role of IL-6 is classically considered less important, while other agents (e.g. B cells, IFN) carry a crucial pathogenetic role. Therefore, if excluding such an important and prevalent condition like RA would have been a strong limitation of our study, its inclusion may hopefully strongly corroborate our findings.

4. The Methods section lacks citations related to the classification of MCTD and SSc. Anti-synthetase syndrome and UCTD are mentioned in the Results, but their classification criteria should be referenced in the Methods section.

Thank you: all missing cr

---

## [Decision Letter · Decision Letter 1]

15 Sep 2024

PONE-D-24-29428R1Panel of serum biomarkers for differential diagnosis of idiopathic interstitial lung disease and interstitial lung disease-secondary to systemic autoimmune rheumatic diseasePLOS ONE

Dear Dr. Dalessandro,

Thank you for submitting your manuscript to PLOS ONE. After careful consideration, we feel that it has merit but does not fully meet PLOS ONE’s publication criteria as it currently stands. Therefore, we invite you to submit a revised version of the manuscript that addresses the points raised during the review process.

The manuscript was much improved by revisions, but one of the reviewers still has suggestions for improved quality of the contents. 

We look forward to receiving your revised manuscript.

Kind regards,

Masataka Kuwana, MD, PhD

Academic Editor

PLOS ONE

Journal Requirements:

Reviewers' comments:

Reviewer's Responses to Questions

**Comments to the Author**

1. If the authors have adequately addressed your comments raised in a previous round of review and you feel that this manuscript is now acceptable for publication, you may indicate that here to bypass the “Comments to the Author” section, enter your conflict of interest statement in the “Confidential to Editor” section, and submit your "Accept" recommendation.

Reviewer #1: All comments have been addressed

Reviewer #2: (No Response)

2. Is the manuscript technically sound, and do the data support the conclusions?

Reviewer #1: Yes

Reviewer #2: Yes

3. Has the statistical analysis been performed appropriately and rigorously? 

Reviewer #1: Yes

Reviewer #2: Yes

4. Have the authors made all data underlying the findings in their manuscript fully available?

Reviewer #1: Yes

Reviewer #2: Yes

5. Is the manuscript presented in an intelligible fashion and written in standard English?

Reviewer #1: Yes

Reviewer #2: Yes

6. Review Comments to the Author

Reviewer #1: (No Response)

Reviewer #2: Comments for Authors

Regarding Major Comment 3, the response provided is insufficient.

The question raised was whether there exists a previously established definition for the term or disease concept of ARD (systemic ARD). For instance, the concept of UCTD has been in use since its proposal by LeRoy et al. (LeRoy EC, Maricq HR, Kahaleh MB. Undifferentiated connective tissue syndromes. Arthritis Rheum. 1980 Jan;23(1):1-9.). Is there an established concept for SARD? If so, a reference should be cited, and the concept should be briefly described. If it is not an established term (i.e., coined by the authors), a brief explanation of the rationale behind using the term "SARD" should be provided.

Regarding Major Comment 9, the response is again insufficient.

If Table 2 presents only statistically significant results, this should be clearly stated in the figure legend. Additionally, “W” should be described as “weight” in the legend.

Regarding Minor Comment 13, Table 1 has not been revised as requested.

The authors' response is also unclear. Please arrange the order of the ARD subgroup diseases in Table 1 as follows: SSc Limited, SSc Diffuse, UCTD, etc.

7. PLOS authors have the option to publish the peer review history of their article (what does this mean?). If published, this will include your full peer review and any attached files.

Reviewer #1: No

Reviewer #2: No

---

## [Author Response · Author response to Decision Letter 1]

16 Sep 2024

Dear Editor,

The authors would like to thank you and appreciate the Reviewer for the constructive comments. We also thank the Reviewer for the effort and time put into the review of the manuscript. The comments are encouraging, and the Reviewer appears to share our judgement that this study and its results are important and worth publication. Each comment has been carefully considered point by point and responded to. Please see below our detailed response to the comments raised, which you can find in italic. Changes provided in the revised manuscript are in red. The manuscript has been revised according to the Reviewer’s suggestions and has been significantly improved. We are hoping that its final version deserves publication in “Plos One”.

Reviewer #2: Comments for Authors

Regarding Major Comment 3, the response provided is insufficient.

The question raised was whether there exists a previously established definition for the term or disease concept of ARD (systemic ARD). For instance, the concept of UCTD has been in use since its proposal by LeRoy et al. (LeRoy EC, Maricq HR, Kahaleh MB. Undifferentiated connective tissue syndromes. Arthritis Rheum. 1980 Jan;23(1):1-9.). Is there an established concept for SARD? If so, a reference should be cited, and the concept should be briefly described. If it is not an established term (i.e., coined by the authors), a brief explanation of the rationale behind using the term "SARD" should be provided.

We are sorry for the unclear answer to your question. Systemic autoimmune rheumatic diseases (SARD) is a term that covers a broad spectrum of clinical conditions of autoimmune aetiology and, in particular, ILD has been reported in association with several SARD, particularly RA, SSc, IIM, CTD and AAV. The term was first proposed in 1995 (Sénecal et al., J Rheumatol) in a cohort of patients displaying anti-SSA/SSB positivity and the most recent, comprehensive, definition was given in a recently published review (Guthridge et al., Nat Med, 2022, PMID 35788174), whose reference has been therefore added to the bibliography of our manuscript. Finally, specifically focusing on lung involvement in rheumatic diseases, the recently published recommendations of the British Society of Rheumatology employ the term SARD for ILD associated with CTD, AAV, IIM, RA etc. (Hannah et al., Rheumatol Adv Pract, 2024)

Regarding Major Comment 9, the response is again insufficient.

If Table 2 presents only statistically significant results, this should be clearly stated in the figure legend. Additionally, “W” should be described as “weight” in the legend.

Thank you for the comment. We included the description of “W” at the top of the column and we improved the caption of table 2 as follows:

“Table 2. Serum markers concentrations in groups of patients stratified according to HRCT findings: the idiopathic ILD group included probable fibrotic NSIP on HRCT (IPF confirmed at multidisciplinary discussion, >65 years old, without UIP confirmation at lung biopsy), definite UIP and definite fibrotic NSIP (UIP confirmed at lung biopsy). For SARD patients, those with RA and SSc displaying a UIP pattern (SARD-UIP) were considered separately from those with NSIP (SARD-NSIP). Abbreviations: SARD, autoimmune rheumatic disease; NSIP, non-specific interstitial pneumonia; UIP, usual interstitial pneumonia; KL-6, krebs von den lungen-6; FER, ferritin; IL-6, interleukin-6; SMRP, soluble mesothelin-related peptide.”

Regarding Minor Comment 13, Table 1 has not been revised as requested.

The authors' response is also unclear. Please arrange the order of the ARD subgroup diseases in Table 1 as follows: SSc Limited, SSc Diffuse, UCTD, etc.

Thank you for the suggestions. We modified the table 1 accordingly.

---

## [Decision Letter · Decision Letter 2]

18 Sep 2024

Panel of serum biomarkers for differential diagnosis of idiopathic interstitial lung disease and interstitial lung disease-secondary to systemic autoimmune rheumatic disease

PONE-D-24-29428R2

Dear Dr. Dalessandro,

We’re pleased to inform you that your manuscript has been judged scientifically suitable for publication and will be formally accepted for publication once it meets all outstanding technical requirements.

Kind regards,

Masataka Kuwana, MD, PhD

Academic Editor

PLOS ONE

Additional Editor Comments (optional):

Reviewers' comments:

Reviewer's Responses to Questions

**Comments to the Author**

1. If the authors have adequately addressed your comments raised in a previous round of review and you feel that this manuscript is now acceptable for publication, you may indicate that here to bypass the “Comments to the Author” section, enter your conflict of interest statement in the “Confidential to Editor” section, and submit your "Accept" recommendation.

Reviewer #2: All comments have been addressed

2. Is the manuscript technically sound, and do the data support the conclusions?

Reviewer #2: Yes

3. Has the statistical analysis been performed appropriately and rigorously? 

Reviewer #2: Yes

4. Have the authors made all data underlying the findings in their manuscript fully available?

Reviewer #2: Yes

5. Is the manuscript presented in an intelligible fashion and written in standard English?

Reviewer #2: Yes

6. Review Comments to the Author

Reviewer #2: "The revised manuscript has been sufficiently improved. I hope this study proposes novel biomarker candidates for SARD-ILD and contributes to the advancement of research in understanding the pathogenesis of SARD-ILD.

7. PLOS authors have the option to publish the peer review history of their article (what does this mean?). If published, this will include your full peer review and any attached files.

Reviewer #2: No

---

## [Editor Report · Acceptance letter]

24 Sep 2024

PONE-D-24-29428R2 

PLOS ONE

Dear Dr. d’Alessandro, 

I'm pleased to inform you that your manuscript has been deemed suitable for publication in PLOS ONE. Congratulations! Your manuscript is now being handed over to our production team.

Kind regards, 

on behalf of

Prof. Masataka Kuwana 

Academic Editor

PLOS ONE